# Small Hydropower in the Baltic States—Current Status and Potential for Future Development

**Egidijus Kasiulis \*, Petras Punys, Algis Kvaraciejus, Antanas Dumbrauskas and Linas Jurevičius**

Institute of Water Resources Engineering, Vytautas Magnus University, 10 Universiteto Str., Akademija, LT-53361 Kaunas, Lithuania; petras.punys@vdu.lt (P.P.); algis.kvaraciejus@vdu.lt (A.K.); antanas.dumbrauskas@vdu.lt (A.D.); linas.jurevicius@vdu.lt (L.J.)

**\*** Correspondence: egidijus.kasiulis@vdu.lt

**Abstract:** Small hydropower (SHP) plays an important role in the Baltic States as a reliable and efficient source of electricity from renewable sources. This study presents the historical development, current status, and possible trends for the future development of SHP in the Baltic States with insights into the legal background and development policies for SHP in Estonia, Latvia and Lithuania. For the assessment of hydraulic structures in the Baltic States, the historical data was used supplemented with data from hydropower associations, the national departments of statistics, electricity transmission systems operators, etc. The currently recommended best practice for SHP development is the utilisation of existing sites with available infrastructure. These include old water mill sites and existing dam sites, which, in the Baltic States, number more than 1500. The majority of these sites have their power potential attributed to micro-hydro (<100 kW). In this study the potential of the hydropower capacity at historic, currently nonpowered dams is evaluated and the distribution of the micro-hydro sites within the EU network of protected areas under Natura 2000 and nationally designated areas is presented. The potential electricity generation from such historic sites in the Baltic States is more than 200 GWh/year.

**Keywords:** small hydropower; current status; potential; historic nonpowered sites; micro-hydro; protected areas; Baltic States

---

## 1. Introduction

Since the adoption of the EU Renewable Energy Directive in 2009, most Member States have experienced significant growth in renewable energy deployment. Renewable energy consumption rose from a 9% share in 2005 to 16.7% in 2015. Consequently, the EU is on track toward the 2020 target of 20%. The "Clean Energy for all Europeans" package adopted in November 2016, included a Renewable Energy legislative proposal that introduced further targets of a minimum of a 27% share of renewable energy consumed in the EU by 2030 [1].

According to the Eurostat data, in 2018, hydropower was Europe's second largest renewable energy resource accounting for 11.8% of the total electricity production from renewable energy in the EU-28 countries, topped only by wind with 12.2%. Hydropower, excluding pumped storage, was a source of renewable electricity production amounting to 363.3 TWh in 2018 in the EU-28 countries. The small hydropower (SHP) share was approximately 15% of the total hydroelectricity generation [2,3].

In 2005, hydropower (small and large) accounted for over 70% of all electricity generated from renewable sources in the EU-27 countries. National renewable energy action plans (NREAPs) correctly predicted that the share would drop to approximately 30% by 2020 due to the high growth rate of other renewables (especially wind, biomass, and solar) [4]. Even if slow, the development of small hydropower in the EU is still expected by 2020. Although the data from different sources is inconsistent,

from 2016, the growth of small hydro is anticipated on average by 1.5 GW of the total capacity by 2020 (Table 1).

**Table 1.** Small hydropower current state and previsions up to 2020 in the EU-28.

|  | 2016 | | | Anticipated by 2020 | | |
|---|---|---|---|---|---|---|
|  | EurObserv'ER [5] | International Center on Small Hydropower [6] | EurObserv'ER [5] | National Renewable Energy Action Plans [4] | European Small Hydropower Association [7] | Green-X Project [8] |
| Total capacity, GW | 14.3 | 15.0 | 15.0 | 16.3 | 17.3 | - |
| Generation, TWh/year | 47.2 | 54.5 | 45.0 | 54.1 | 59.7 | 68.4 |

The Baltic States—Lithuania, Latvia, and Estonia—are often referred to as the Baltic sisters due to their similar historical and cultural patterns. These countries regained their independence in 1990 and joined the EU in 2004. Their similarity is also reflected in their history, development, and current situation of small hydropower.

The Baltic States are relatively low-lying and flat countries that are situated along the southeastern shore of the Baltic Sea (Figure 1a). The climate is humid, continental, and temperate owing to the maritime influence of the Baltic Sea. The average elevation of Estonia is only about 50 m above sea level. Although Estonia has numerous rivers, they do not result in significant hydroelectric potential. Approximately 98% of Latvia's area lies under 200 m of elevation. Lithuania is also flat, except for a large number of undulating hills in the western uplands and eastern highlands; however, these are no higher than 290 m. About 57% of Lithuania lies below 100 m above the sea level and only 2.5% lies above 200 m (Figure 1b).

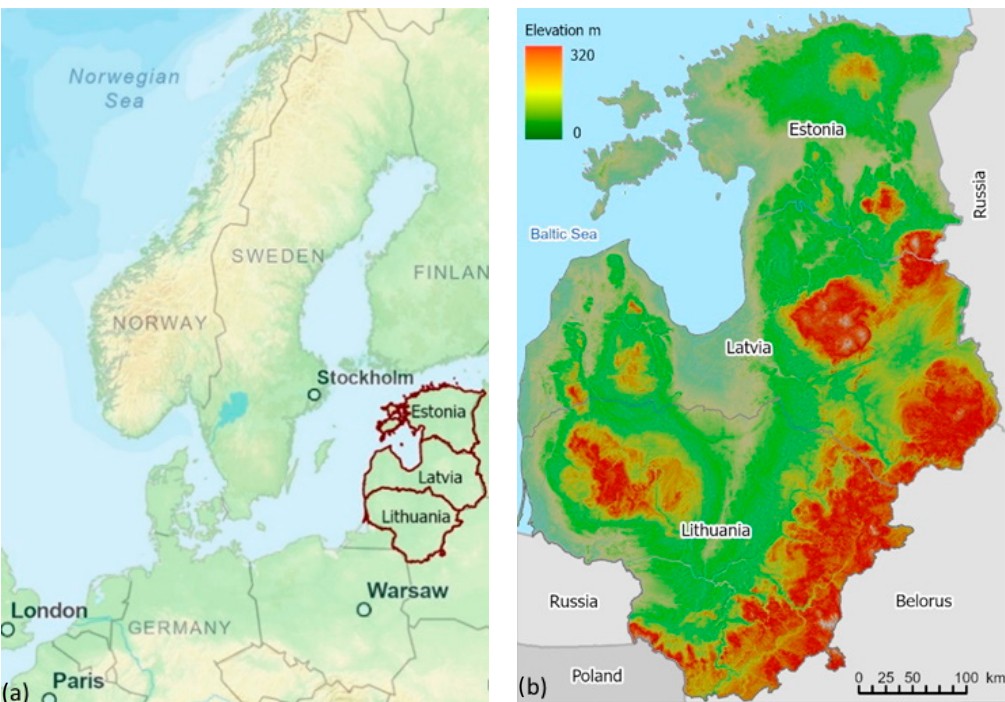

**Figure 1.** (**a**) Location of the Baltic States in the Baltic Sea region (Source: © Esri, open street map contributors), and (**b**) elevation map of the Baltic States (Source: © Esri, © USGS SRTM, open street map contributors).

The development of hydropower in the Baltic States is becoming increasingly challenging as it is commonly known that hydropower development, associated with river damming, might result in water stream physical modifications and, consequently, affect the habitats of aquatic species. As there

are many technologies involved, hydropower plants can affect the environment in many ways and on different levels [9–12]. Therefore, there is a need to look at each case individually. Although there can be common features for some species/habitats or types of technologies, the effect of hydropower plants on particular sites, including the species and habitats, are always site specific [13–15]. Distinguishing between significant and insignificant effects is a challenging task. The assessment of significance should also be considered over an appropriate geographical scale, e.g., a basin scale [16].

Stream physical modifications and habitat alterations can vary from mild for some types of hydropower plants to very heavy, particularly for large reservoir (storage) hydropower plants. However, even small hydropower schemes can have significant adverse impacts on water habitats, particularly if they are inappropriately designed and located or are a part of the cumulative impact of multiple barriers and flow disturbance in a single catchment [17].

There is no difference in the legal approach in the Baltic States between small and large hydropower although the technical approach and specific assessment of impacts can be different. This is mainly because there are only four large hydropower plants in the Baltic States (excluding pumped storage power plants) and no further development of a large hydropower is foreseen. Therefore, all laws regarding an environmental approach to hydropower development now mainly concern small hydropower.

In the Baltic States, the following structure of protected areas, where hydropower development is prohibited, can be identified:

- nationally designated protected areas under the Laws of Protected areas.
- the EU Natura 2000 protected areas network established under the EU Habitats (92/43/EEC) and Birds (2009/147/EC) directives that form the cornerstones of the EU's biodiversity policy.
- rivers and their stretches exempt from damming or so-called "no-go areas" for hydropower development

The first two types of protected areas can be called conventional ones. The last entity, generally listing rivers and their stretches protected from hydropower was passed by a governmental order initiated by the Environmental authorities and is not subject to the national laws of protected areas. The lists of "no go areas" overlap with nationally designated areas and the Natura 2000 network and extend far beyond them. There is neither any flexibility nor strategic planning as a blanket ban has been imposed for these rivers. In contrast, under certain conditions, exemptions of a plan or project for sites designated in the EU Natura 2000 network can be applied. According to Article 6, this network is not designed to be a "no go area", and new developments are not excluded a priori [18].

In Estonia and Latvia, there are 125 and 214 rivers and their stretches that are exempt from further dam construction. In Estonia, the list of "no go areas" is intended for the protection of salmonids, their spawning areas, and habitats [19]. In Latvia, these designations aim to protect all fish resources [20].

In Lithuania, a similar environmental law was in force from 2004 to 2019 [21]. In this case, 170 rivers and their stretches were named as "environmentally and culturally sensitive rivers". This blanked ban in Lithuania was briefly reviewed in [22]. Currently, the blanket ban is lifted, and the amendment of the Water Law of the Republic of Lithuania, which is in force, prohibits dam constructions in protected areas and disallows such constructions if they do not ensure good water status requirements according to Directive 2000/60/EB. The main difference is that this amendment in Lithuania also prohibits dam construction in the largest river of Lithuania—Neman, while the largest rivers of Latvia and Estonia—Daugava and Narva—are not protected as "no go areas".

The result of such extensive unconventional structures of the protected rivers is that the Baltic States have one of the strictest environmental laws in the whole EU that strongly impedes hydropower development. The construction of new dams is also highly dubious. On the other hand, the currently recommended best practice for hydropower development is not the construction of new dams. In the Baltic States there are hundreds of existing sites, old watermills and dams, built during the Soviet era, that could be retrofitted with hydroturbines and generate electricity [23,24].

The aim of this study was to assess issues for small hydropower development in the lowland Baltic States with additional detailed focus on historic and currently nonpowered micro-hydro sites. The overall objectives of the study are:

- to present a brief history and overview of the current status of small hydropower in the Baltic States,
- to analyse environmental laws and support schemes and national renewable energy strategies that affect the future development of small hydropower in the Baltic States, and
- to evaluate the possible capacity and electricity generation at historic and currently nonpowered dam sites in the Baltic States.

## 2. Materials and Methods

The official small hydropower classification in terms of capacity in Estonia, Latvia, and Lithuania is below 10 MW. In this paper, a more detailed breakdown in capacity (P, kW, or MW) is used as follows: micro hydro—up to 100 kW; mini hydro—above 100 kW and below 1 MW; and small hydro—above 1 MW and below 10 MW [25].

Data collected during the Restor Hydro project was used in this paper as the initial historical data on hydraulic structures in the Baltic States. These data were supplemented with data from the Lithuanian and Latvian hydropower associations, national statistics departments, electricity transmission system operators, etc. The methodology used in the Restor Hydro project is described in detail in [26]. The available data from the Restor Hydro project on hydraulic structures in the Baltic States were location, historical use (watermill or dam); restorable conditions (degraded, moderate or advanced); expected mean flow and expected head. The latter two were used for calculation of the potential annual electricity generation for these sites.

The overall potential of hydropower was not reassessed in these countries. Only the potential of historical watermills sites and existing dams that were not previously used for electricity production was distinguished in the Baltic States.

In this paper, to describe such existing water infrastructure, supplementary terms of nonpowered dams will be used. The potential capacity and possible electricity generation was estimated for historic and nonpowered dams. The proportion of the potential of such nonpowered dams in the remaining hydropower potential in the Baltic States was evaluated. The potential annual electricity generation (E, kWh/year) from a historic and nonpowered dam sites was calculated using general formula:

$$E = PT = 8QHT, \tag{1}$$

where $P$ is the capacity (kW), $T$ is the number of operating hours at installed capacity (on average $T = 4000$ h/year), 8 is the coefficient that is the product of the specific weight of water ($\gamma = 9.81$ kN/m$^3$) and the overall plant efficiency ($\eta = 0.6$–$0.9$), $Q$ is the flow (m$^3$/s), and $H$ is the head (m).

The multiplier of 8 results in the efficiency magnitude of 0.81 which can be described as "losses". These losses may include, but not be limited to head-loss in penstocks or forebays, water wheel efficiency and friction losses, transmission and generator loss and electricity conversion or conditioning for connecting to the local or commercial grid. If a site is problematic and maximum efficiencies are likely not attainable, this multiplier can be reduced from 8 to 7 or even 6, resulting in a more conservative design capacity estimate. Then the overall efficiency of a hydropower plant is between 0.71 and 0.61, respectively.

The environmental regulations for adding power to the existing historic site infrastructure located in the Natura 2000 areas and other environmentally sensitive areas differ from country to country—from a relatively liberal approach to a blanket ban [22]. This crucial issue for developing hydro schemes was considered in this paper.

Available spatial information to screen out sites in protected areas is one of the most important issues in developing hydro schemes. Two of the largest European inventories of these, the Natura

2000 (Network of Nature Protection Areas in the territory of the European Union) [27] and Nationally Designated areas (CDDA—Common Database on Designated Areas) [28], were considered.

Advanced GIS tools with simple data processing procedures (overlapping and attributed to the tabulation methods available in the ArcGIS Pro toolset) were used. The GIS data layers of locations of historic hydro sites, the river network, nationally designated areas, and Natura 2000 sites were included in the spatial analysis. To produce unified GIS layers, the national coordinate systems were transformed to the EPSG:3035 coordinate reference system.

For the calculation of the contribution of SHP plants for the purposes of Directive 2009/28/EC, the effects of the climatic variation (e.g., wet or dry years) were smoothed through the use of a normalisation rule. Due to the relatively short series of SHP power generation in Estonia, the normalised generation can be regarded as a preliminary estimate.

## 3. Results

### 3.1. Short Overview of Electricity Sector in the Baltic States

#### 3.1.1. Estonia

The electricity sector in Estonia is dominated by oil shale power plants. The shift toward renewables is substantial, and oil shale is a major owned resource in the country, which makes Estonia self-sufficient and energy independent. The installed capacity of renewables is gradually increasing. In 2008, the share of oil shale in the total electricity generation was a colossal 93.6%, followed by 2.8% from natural gas, 1.9% from oil shale gas, and then by only 1.2% from all renewables [29]. At present, the share of oil shale in the total electricity generation decreased to 75.8%, and the share of renewables increased to 15.7% [30]. The evolution of the distribution of electricity generation from different renewable energy sources in Estonia can be observed in Figure 2.

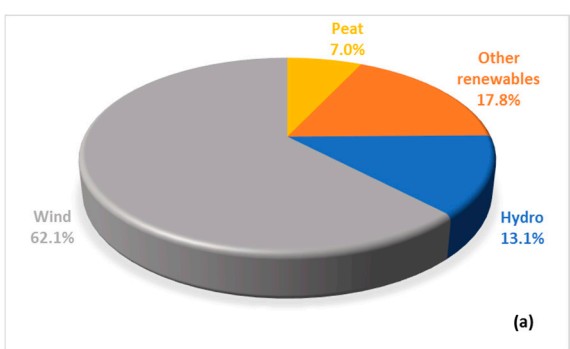 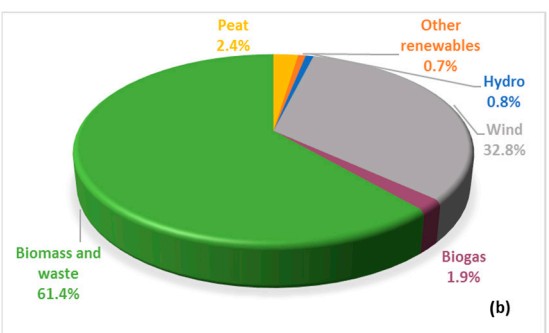

**Figure 2.** The distribution of electricity generation from different renewable energy sources in Estonia: (**a**) 2008 and (**b**) 2018 [30].

As can be seen from Figure 2, the main renewable sources for electricity generation in 2008 in Estonia were wind, hydropower, and peat, while in 2018 they were biomass, waste, wind, peat, and biogas. Several conclusions can be drawn from this figure. First, during this period the wind remained an important source of electricity, while the share of hydropower became negligible. Secondly, the biomass sector grew into the largest contributor to renewable energy generation. This can be explained by the fact that part of oil shale plants was retrofitted to use the biomass as fuel as well. Thirdly, as the northernmost country of the three Baltic States, Estonia, is not developing in the solar energy sector. The generation of electricity from renewable sources in Estonia from 2008 to 2018 increased nine times: from 214 GWh in 2008 to 1940 GWh in 2018 [30].

### 3.1.2. Latvia

The electricity sector in Latvia is dominated by large hydropower plants and cogeneration plants. The most recent plant from the three large hydropower plants' cascade on the largest Latvian river, Daugava, was commissioned in 1979. This means that Latvia, according to 2018 data, is the country in the European Union with the third highest share of energy coming from renewable sources (40.3%). The first two are Sweden (54.6%) and Finland (41.2%) [2]. The evolution of the distribution of renewable energy from different sources in Latvia is presented in Figure 3.

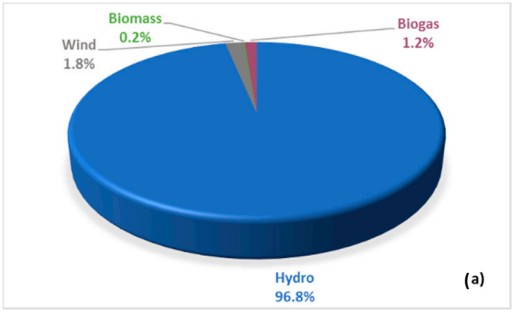 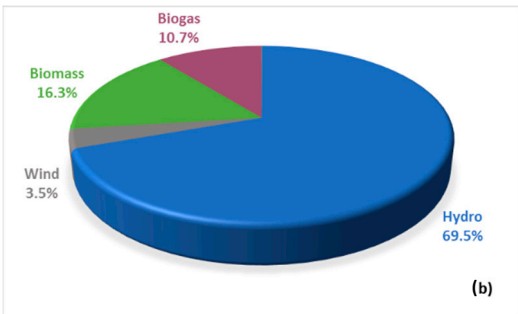

**Figure 3.** The distribution of electricity generation from different renewable energy sources in Latvia: (**a**) 2008 and (**b**) 2018 [31].

The boost of utilisation of biomass and biogas in Latvia can be seen in Figure 3. This can be explained by growth in the capacity of cogeneration plants fuelled by these two types of renewable energy sources [32]. The share of electricity coming from hydropower will constantly decrease as there are no plans for the construction of new large hydropower plants in Latvia. The growth in wind energy was observed; in 2018, the first GWh of electricity generated from solar power plants was registered in Latvia. However, the increase in the total generation of electricity from renewable sources in Latvia from 2008 until 2018 is not as inspiring: 3213 GWh in 2008 and 3499 GWh in 2018 [31].

### 3.1.3. Lithuania

Over one night in 2009, after the closure of the Ignalina nuclear power plant, Lithuania changed from an electricity-exporting country to an electricity importing country. At that time, two large hydropower sites—the Kaunas hydropower plant (installed capacity 100.8 MW) and Kruonis pumped storage plant (installed capacity 900 MW)—were operating. Afterward, first to reduce the amount of imported fossil fuel and then to meet EU requirements, Lithuania began the rapid development of renewables. This led to the fact that currently the dominant sources of generated electricity in Lithuania are wind and hydropower. The evolution of the distribution of renewable energy from different sources in Lithuania is presented in Figure 4.

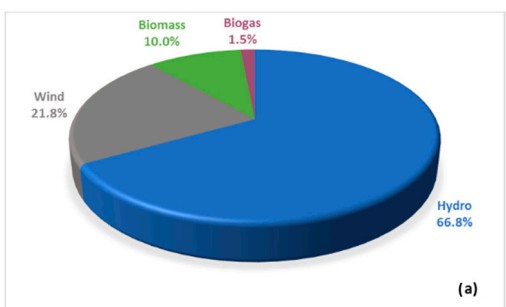 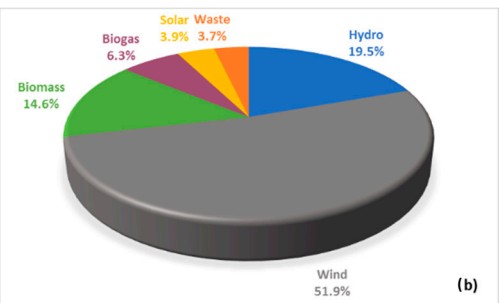

**Figure 4.** The distribution of electricity generation from different renewable energy sources in Lithuania: (**a**) 2008 and (**b**) 2018 [33].

In Figure 4, the electricity generated at the Kruonis pumped storage plant was not included in the electricity generated from hydropower. This plant annually generates approximately 600 GWh [34], and, if this value is included, the amount of electricity generated from hydropower would increase more than twice. From Figure 4, Lithuania is the fastest developer in all three Baltic States of wind and solar power. In 2013, the first 10 MW capacity waste-to-energy plant was commissioned in Lithuania, which, in 2018, generated approximately the same amount of electricity as all solar photovoltaic installations. The capacity of all solar photovoltaic installations in Lithuania in 2018 was 82 MW. The generation of electricity from renewable sources in Lithuania from 2008 to 2018 increased almost four times: from 602 GWh in 2008 to 2205 GWh in 2018 [33].

### 3.1.4. Toward a Renewable Future

The growth in electricity generation from renewable energy sources aided all the Baltic States to reach their targets ahead of time regarding the share of energy from renewable sources in the final energy consumption set for 2020. Lithuania and Estonia reached their target in 2015 and Latvia in 2018 [2]. The next step will be new targets set for 2030, and the national renewable energy action plans (NREAPs) will be followed by national energy and climate plans (NECPs). The NECPs goals for Estonia are a 42% share of renewable energy in the final consumption of energy and a 40% share of renewable electricity. For Latvia, these goals are 50% and 60% and, for Lithuania, 45% and 45%, respectively [35].

The NECP considerations for hydropower development in the Baltic States is almost negligible. In Estonia's NECP, there are plans for the construction of two new pumped storage plants with a total capacity of 550 MW, while in Latvia's NECP, there are only plans to update the spillway of the Plavinas hydropower plant—the largest hydropower plant in the Baltic States with an installed capacity of 894 MW. This would increase the dam safety and provide better opportunities for flood protection. The only small window for SHP development according to the NECP has Lithuania; the total installed capacity of all SHP plants in Lithuania by 2030 will have to reach 27 MW. Unfortunately, the total installed capacity of SHP plants in Lithuania in 2017 was 26.9 MW.

### 3.2. History and the Current Status of Small Hydropower in the Baltic States

The first written evidence of the existence of the watermills in the Baltic States dates as early as the 11th century. Together with windmills, they are the oldest industrial objects in these countries. The power of water via the waterwheels was used to operate grain mills, sawmills, wool carding mills, paper mills, and other industrial objects. The first road systems were linked with watermills, as dams were also used as bridges [36].

It is difficult to determine the exact number of watermills that were operating at the end of the 19th century, when the first hydropower plants were built in Lithuania, Latvia, and Estonia. It is known that, in Lithuania and Latvia, there were approximately 640 and 700 watermills. The reconstruction of these watermills, replacing the waterwheel with the turbine and connecting it to the generator, thereby, transforming the watermills into small or micro hydropower plants, stared the era of hydropower in these three countries.

Before World War II, small hydropower became one of the main sources of electricity in the rural areas, and this forced the development of the local electricity grids. There were 96 small hydropower plants in Lithuania in 1935 with a total installed capacity of 1.9 MW [37]. In Latvia in 1926, there were 26 SHP plants with a total installed capacity of 1.5 MW [38]. At that time, Estonia had the most developed network of hydropower plants, because, in 1940, there were 921 operational waterwheels and hydroturbines, while the total installed capacity of the hydropower plants was 9.3 MW [39].

During World War II, the majority of SHP plants were destroyed; however, after the war, the restoration work began immediately together with the construction of new plants. The status of small hydropower grew together with the number of plants. In 1949 in Latvia, there were 60 SHP plants

with the installed capacity of 5.8 MW [40], while, in Lithuania in 1958, there were already 104 SHP plants producing 19 GWh annually [37].

The decline of small hydropower started in 1954 with the beginning of construction of the joint electricity grid. After the construction of large hydro, nuclear, and thermal power plants, the SHP plants became unprofitable and were eliminated from the grid. By 1977, there were no SHP plants operating in Latvia [38] and Estonia [39], while, in Lithuania in 1981, there were only 13 still operating [37].

Small hydropower was rediscovered in Lithuania, Latvia, and Estonia after 1990 when all three Baltic States regained independence. Many old SHP plants and watermill sites were rebuilt and began to produce electricity once again. The growth in the number of SHP plants in the Baltic States from 1990 is presented in Figure 5.

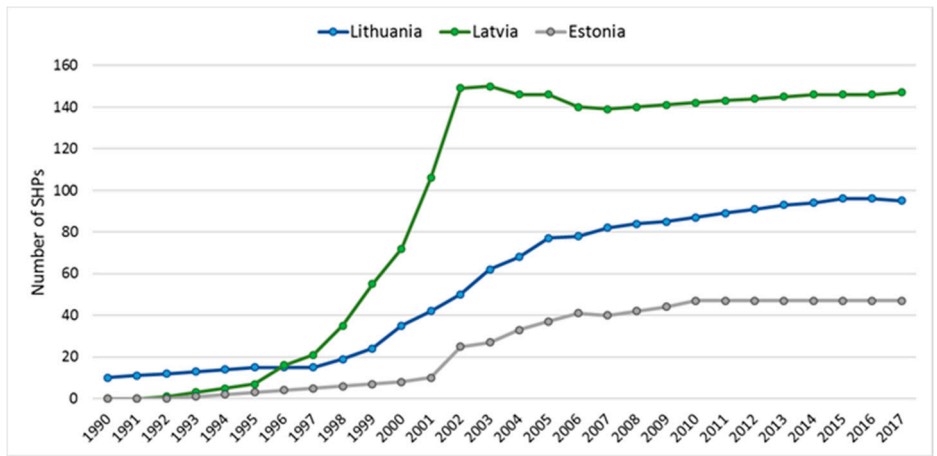

**Figure 5.** The growth in the number of small hydropower plants in the Baltic States. SHP: small hydropower.

The year 1990 started with Lithuania with 10 SHP plants and Latvia and Estonia with none. Six years later in Latvia, there were already more SHP plants than in Lithuania, and, as can be seen from Figure 5. Latvia became and still is the biggest developer of small hydropower in the Baltic States. In 2017 in Latvia, there were 147 small hydropower plants, while, in Lithuania, there were 95 and in Estonia, 47.

The impressive growth of the number of SHP plants in Latvia until 2002 was related to favourable state support policies for all renewables due to the lack of electricity generators. At that time in Latvia, approximately 50% of the consumed electricity in the country was generated locally. Lithuania still had its Ignalina nuclear power plant, and Estonia had its oil shale power plants, which meant that surplus capacities were available in these countries. Therefore, there was no need for such exclusive support. The sharp decline in the development of small hydropower in all three Baltic States starting from 2002 is clearly visible in Figure 5. This was due to the stringent adopted environmental laws that stopped further development.

Interestingly, a large number of SHP plants in Latvia did not respond in terms of the actual electricity generation. Although, in Lithuania, there are nearly 1.5-times less SHP plants and, by the same amount, less total volume of water in the rivers than in Latvia, the electricity generation at the SHP plants is much higher (Figure 6). During the period from 1990 to 2017 at SHP plants in Lithuania, a total of 1437 GWh, in Latvia a total of 1180 GWh and in Estonia a total of 429 GWh of electricity was generated. This also reflects in normalised electricity generation evaluations for the same period (Figure 6).

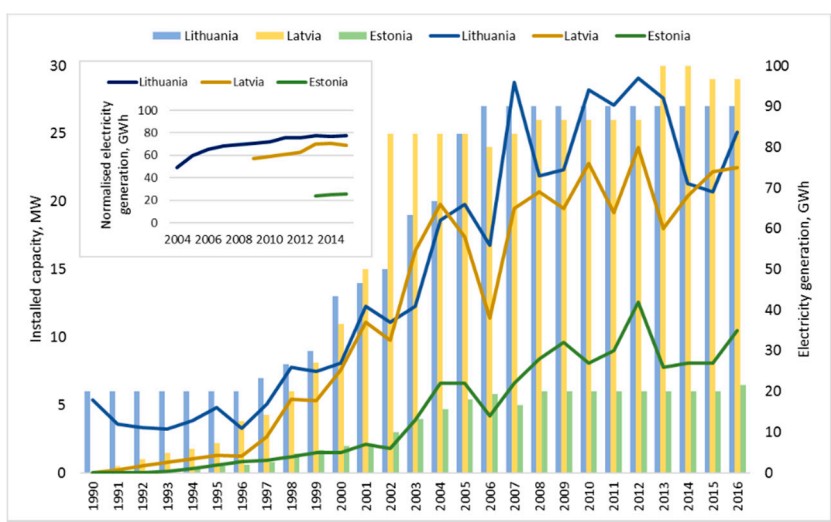

**Figure 6.** The installed capacity and electricity generation of small hydropower plants in the Baltic States for the period from 1990 to 2016.

As it was mentioned, the Baltic States reached their targets of the share of energy from renewable sources in the final energy consumption set for 2020 ahead of schedule. On the other hand, these targets were reached with minimal input from the development of hydropower. From the hydropower perspective, the targets of the NREAPs looked discouraging (Figure 7). Estonia did not foresee the development of small hydropower after 2011 and still the targets for small hydropower were never reached. Lithuania's target was the rapid development of small hydropower, when, in reality, those targets were also never reached, and, according to the 2016 figures, the lag from the target was quite large—6 MW. Only Latvia, not only exceeded its target for small hydropower already in 2013 but also, by doing this, at the same year reached the target for 2020 regarding the installed capacity of SHP plants. Unfortunately, the foreseen growth in the installed capacity in Latvia for SHP plants for the period from 2005 to 2020 was only 3 MW, when, for example in Lithuania for the same period, the same growth was foreseen as 13 MW.

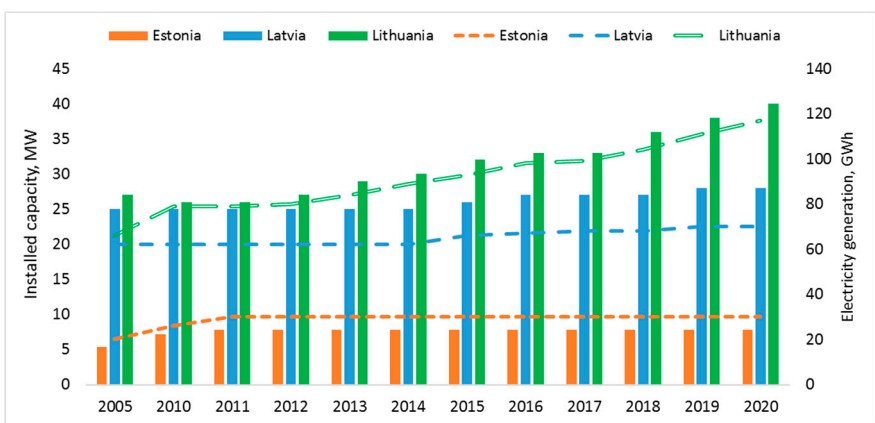

**Figure 7.** The targets of the national renewable energy action plans (NREAPs) for small hydropower development in the Baltic States.

For comparison, the installed capacity and electricity generation of other renewables are juxtaposed with the NREAP goals in Table 2. The renewables that are presented in Table 2 are wind power (land and sea), sun power (photovoltaic), and biomass (solid).

**Table 2.** The installed capacity and electricity generation from renewable energy sources with comparison to the national renewable energy action plan (NREAP) targets in Estonia, Latvia, and Lithuania.

| | | 2012 | | 2014 | | 2016 | | 2018 | |
|---|---|---|---|---|---|---|---|---|---|
| | | MW | GWh | MW | GWh | MW | GWh | MW | GWh |
| **Estonia** | Wind (installed) | 266 | 434 | 334 | 604 | 304 | 594 | 306 | 636 |
| | Wind (NREAP goal) | 311 | 432 | 400 | 855 | 500 | 974 | 550 | 1320 |
| **Latvia** | Wind (installed) | 59 | 114 | 69 | 139 | 70 | 128 | 78 | 122 |
| | Wind (NREAP goal) | 49 | 100 | 80 | 175 | 137 | 300 | 235 | 517 |
| | Solar (installed) | 0 | 0 | 0 | 0 | 0 | 0 | 2 | 1 |
| | Solar (NREAP goal) | 1 | 1 | 1 | 1 | 1 | 2 | 2 | 4 |
| | Biomass (installed) | 23 | 65 | 63 | 319 | 81 | 427 | 97 | 570 |
| | Biomass (NREAP goal) | 8 | 49 | 34 | 198 | 55 | 321 | 76 | 447 |
| **Lithuania** | Wind (installed) | 275 | 540 | 288 | 639 | 509 | 1136 | 533 | 1144 |
| | Wind (NREAP goal) | 250 | 563 | 350 | 813 | 500 | 1111 | 500 | 1250 |
| | Solar (installed) | 7 | 2 | 69 | 73 | 70 | 67 | 82 | 87 |
| | Solar (NREAP goal) | 3 | 3 | 8 | 9 | 10 | 15 | 10 | 15 |
| | Biomass (installed) | 38 | 176 | 48 | 292 | 48 | 262 | 51 | 322 |
| | Biomass (NREAP goal) | 40 | 161 | 98 | 416 | 135 | 626 | 162 | 810 |

Estonia in Table 2 is represented only by wind power. This is because Estonia has no target in its NREAP for solar power and, at the same time, no solar power plants. Estonia, in its NREAP, also does not show goals for the installed capacity of biomass power plants, only goals for electricity generation using biomass. The goal in 2018 was to generate 346 GWh of electricity using biomass in Estonia, yet the development exceeded expectations, and the actual generation was steadily increased. In 2018, the electricity generation using biomass and waste was more than three-times higher. Perhaps this, and the fact that Estonia reached the target for the share of energy from renewable sources in final energy consumption set for 2020 in 2015, affected the suspension of wind power growth that can be seen in Table 2. It is clear that Estonia will not reach the NREAP target for wind power. Estonia does not currently have any installed wind power capacity in the sea, although it was foreseen in the NREAP.

Latvia, according to the NREAP prevision, began the development of sun power, although currently the electricity generated from this type of power plant does not reach the planned values (Table 2). As with Estonia, Latvia targeted wind power plants installations in the sea, but did not manage to install any, consequently, falling behind its target for wind power. The only reached and exceeded target, except for small hydro, which was foreseen in NREAP for Latvia was the installed capacity and electricity generation from biomass power plants (Table 2).

In contrast to the other Baltic States, Lithuania reached and exceeded its NREAP targets for the installed capacity from sun and wind power but did not reach the target foreseen for power plants that use biomass (Table 2). All of Lithuania's wind power plants are on land. On the other hand, offshore wind power installations were not foreseen in Lithuania's NREAP.

The important aspect is that, regardless of reaching the targets of the NREAPs or not, in Estonia, Latvia, and Lithuania, growth in the installed capacity and electricity generation of other renewables, except for hydropower, is apparent. The development of hydropower is suspended through strict environmental-oriented laws. However, even bearing in mind that the Baltic States are lowland countries, a small hydropower there is a more favourable form of renewable energy than wind or solar energy.

Solar and wind power are commonly recognised as sources of intermittent electricity. Thus, the amount of available electricity is directly dependent on the availability of the source. Therefore, there will always be a favourable years or unfavourable years for wind or solar power. On the other hand, the recurring dry and wet years directly influence the availability of water for hydroturbines. This corresponds to the variations in the capacity factor values of all three renewable energy sources in the Baltic States. The available annual national statistics [30,31,33] revealed that, during favourable years, the capacity factor for wind energy in Lithuania reached 30%, in Latvia, 23%, and in Estonia,

28%. During wet years, the small hydropower capacity factor in Lithuania and Latvia reached 45%, while in Estonia, 62%. The meaningful statistics for solar energy could be calculated only in Lithuania. There during favourable years, the capacity factor for this renewable energy source reached only 12%.

*3.3. Existing Potential and Support Schemes*

3.3.1. Existing Potential for Small Hydropower

The small hydropower electricity generation potential in the Baltic States, estimated during the Stream Map project, is provided in Table 3 [22,41]. Available technological advancements and existing governmental incentives reduce the gross technical potential to technically feasible and economically feasible potentials, respectively. It is commonly known that the economically feasible potential is less stable than the technically feasible due to the changes in governmental incentives. The environmentally compliant potential, i.e., the potential that excludes the potential that is incompatible with the legal requirements (environmentally protected areas), is the highest in Latvia. The remaining potential, which was estimated excluding the potential of developed sites considering the latest available electricity generation data (from 2017), is also in Latvia.

**Table 3.** The small hydropower electricity generation potential in the Baltic States.

| Potential, GWh/Year | Estonia | Latvia | Lithuania |
|---|---|---|---|
| Gross theoretical | n/a | 1160 | 2093 |
| Technically feasible | 85 | 730 | 853 |
| Economically feasible | 70 | 400 | 617 |
| Environmentally compliant | 50 | 334 | 287 |
| Remaining | 24 | 224 | 180 |

The values, presented in Table 3, considering the annual variations in power generation, coincide rather well with the values that were published in the World Small Hydropower Development Report [3]. In this report, the potential capacity for Estonia was specified as 10 MW. The potential capacity and electricity generation for Latvia were 75 MW and 220 GWh/year, and, for Lithuania, were 52 MW and 126 GWh/year. Thus, a considerable part of the available capacity of small hydropower remains untapped.

3.3.2. Support Schemes for Renewable Energy Producers

In Estonia, renewable energy producers receive subsidies according to the Electricity Market Act. The main grid operator Elering determines the subsidy rate and publishes it every year on the first of December for every calendar year. According to this support scheme, small capacity producers in hydropower, solar, wind, and biomass in cogeneration mode in 2020 are paid the same support fee of 0.0537 Euro/kWh [42]. To further promote the development of renewables, the Estonian Ministry of Economy and Infrastructure in 2020 launched the first small-scale renewable energy reverse auction seeking to procure 5 GWh of renewable energy. The lowest bidders will sign a contract for 12 years. Following reverse auctions are foreseen; therefore, this new support scheme proved to be successful.

In Latvia, renewable electricity generation is stimulated through a complex support system based on a feed-in tariff. The support level for the generation of electricity from renewable energy sources depends on the type of energy source used, the installed capacity of the plant, the number of working hours, and the natural gas sales price. As the gas prices rise, this causes an increase in the support intensity and a rise in the number of supported electricity generators. On the other hand, such a support scheme also causes an increase in the prices of electricity for the final consumers. Therefore, the existing state support mechanisms for energy production from renewable energy sources are being assessed and revised and are currently suspended for new installations [43]. A new support scheme will likely be implemented in the near future on the basis of an approved NECP.

The support scheme for electricity generation from renewable energy sources in Lithuania can be called a sliding feed-in premium. The monthly average market price of electricity, determined by National Energy Regulatory Council, is applied to the producers of electricity from renewable energy sources via the support mechanism through Public Service Obligations (PSO). From the budget of PSO, the producer receives a payment for the difference between the fixed feed-in-tariffs for electricity from the renewable energy sources and the price of the sold electricity. This means that the level of support from the state depends on the electricity market conditions.

This support tariff is guaranteed for 12 years starting from 2011. As of November 2018, the fixed tariff for small hydropower was 0.059 Eur/akWh [44]. For new SHP plants, a fixed tariff and promotional quotas are distributed via reverse auctions. Unfortunately, the promotion quotas are currently exhausted in Lithuania, and the current fixed tariffs for new power plants are no longer applicable. Thus, the situation is similar to the one in Latvia.

Therefore, in theory, all three Baltic States are still supporting small hydropower as a renewable energy source. However, it is very likely that, in the future, small hydropower will have to compete for further support with other renewables in reverse auctions.

### 3.4. Historic and Currently Nonpowered Dam Sites in the Baltic States

#### 3.4.1. Number of Sites

The initial total number of historic and nonpowered dam sites used in this study was 1539 (Table 4). If such sites would be retrofitted with hydroturbines, generally they would be counted as small hydropower sites. Yet, in this paper, we used a more detailed breakdown in capacity, and, as will be proven later, the vast majority of such sites can be classified as micro-hydro sites. Therefore, this description will be further used in this paper.

**Table 4.** Assessment of the historic and nonpowered dam site records in the Baltic States used in the study.

| Country | Number of Sites Used in This Study | | | Density of Micro-Hydro Sites (Number Per 1000 km$^2$) | | |
|---|---|---|---|---|---|---|
| | Total | Watermills | Dams | Total | Watermills | Dams |
| Estonia | 347 | 254 | 93 | 7.7 | 5.6 | 2.1 |
| Latvia | 453 | 367 | 86 | 7.0 | 5.7 | 1.3 |
| Lithuania | 739 | 214 | 524 | 11.3 | 3.3 | 8.0 |
| Total | 1539 | 835 | 703 | 8.8 | 4.8 | 4.0 |

In terms of the quantity, historic watermills represent almost 54% (835 sites) of all micro-hydro sites all over the Baltic region. Lithuania, with a total of 739 micro-hydro locations, is a clear leader among the Baltic countries and represents 48% of the total number in the Baltic States. Table 4 summarises the data used for further analysis.

The total average of the density of micro-hydro sites in the Baltic region was 8.8 per 1000 km$^2$. The highest density of micro-hydro sites was identified in Lithuania, followed by Estonia and then Latvia (Table 4).

The historic watermills prevailed in Estonia (254 sites) and Latvia (367 sites) and varied from 73% to 81% of all recorded micro-hydro sites existing in the country. The largest amount of nonpowered dam sites was identified in Lithuania (524) and represented almost 71% of all micro-hydro locations in the country. A small fraction of micro-hydro sites (12 unknown sites) was not identified as watermills or dams and was not used in the analysis.

#### 3.4.2. Potential Capacity and Electricity Generation

Old watermill sites were typically of low height (an average of 2 to 3 m). Large reservoirs were not created also; therefore, people could construct such mills even in smaller rivers and streams. This is

clearly seen in the potential capacity of old watermill sites in the Baltic States (Figure 8). In Figure 8 all watermill sites used in this study are arranged in descending order according to their potential capacity.

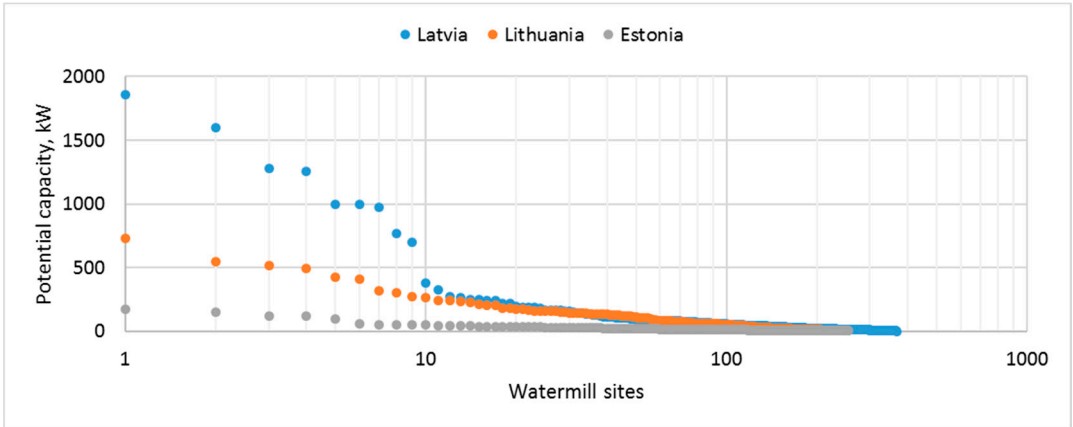

**Figure 8.** The potential capacity of watermill sites in the Baltic States (note the logarithmic *x*-axis).

This is a clear indication of why a more detailed breakdown in capacity was needed. A total of 74% of all old watermill sites' potential capacity were attributed to micro-hydro in Lithuania. In Latvia and in Estonia, this number is even higher—87% and 98%, respectively. Only four such sites were attributed to small hydro in Latvia, and none of the sites were attributed to small hydro in Estonia and Lithuania. The total potential capacity of all old watermill sites in Latvia, Lithuania, and Estonia was 26,281, 16,538, and 3558 kW, respectively. The total potential electricity generation in Latvia was 105.1 GWh/year; in Lithuania, 66.3 GWh/year; and in Estonia, 14.2 GWh/year.

A similar situation was found for nonpowered dam sites. The potential capacity of such sites in the Baltic States is presented in Figure 9. Similarly, as in Figure 8, in Figure 9 all nonpowered dam sites used in this study are arranged in descending order according to their potential capacity.

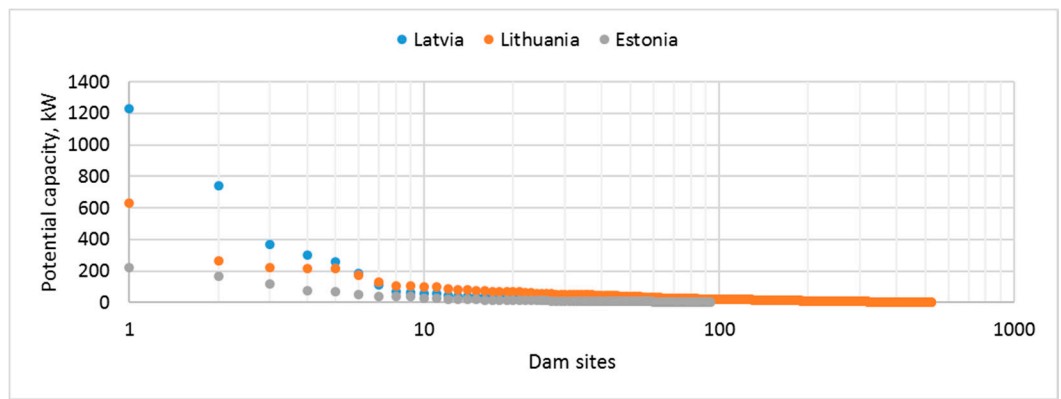

**Figure 9.** The potential capacity of existing dams in the Baltic States (note the logarithmic *x*-axis).

The number of dam sites that were attributed to micro-hydro is even higher—98% in Lithuania, 97% in Estonia, and 92% in Latvia. Only one dam site was attributed to small hydro across three Baltic States, and this site is located in Latvia. The total potential capacity of all existing dam sites in Lithuania, Latvia, and Estonia was 8963, 4371 and 1513 kW, respectively. The total potential electricity generation in Lithuania was 35.9 GWh/year; in Latvia, 17.5 GWh/year; and in Estonia, 6.1 GWh/year.

### 3.4.3. Sites in Environmentally Sensitive Areas

Environmental policy documents recommend first to upgrade existing hydropower plants or use existing in-stream structures—dams, weirs, etc.—before proceeding with new developments [45,46].

This can be viewed as a wise approach, as many of the costs and environmental impacts during dam construction have already been incurred at these sites and may not be significantly increased by the incorporation of new energy production facilities. A possibility of using existing dams and old water mill sites could be one of the means for a sustainable hydropower development. Still, a considerable drawback for this idea is that a large number of historic sites are located in environmentally sensitive areas. More than 51% (428 sites) of all watermills in the Baltic States and analysed in this study are in restricted areas, meaning that they are in nationally protected areas, Natura 2000 areas, or are located in river stretches that are listed as under restriction ("no go areas") (Figure 10).

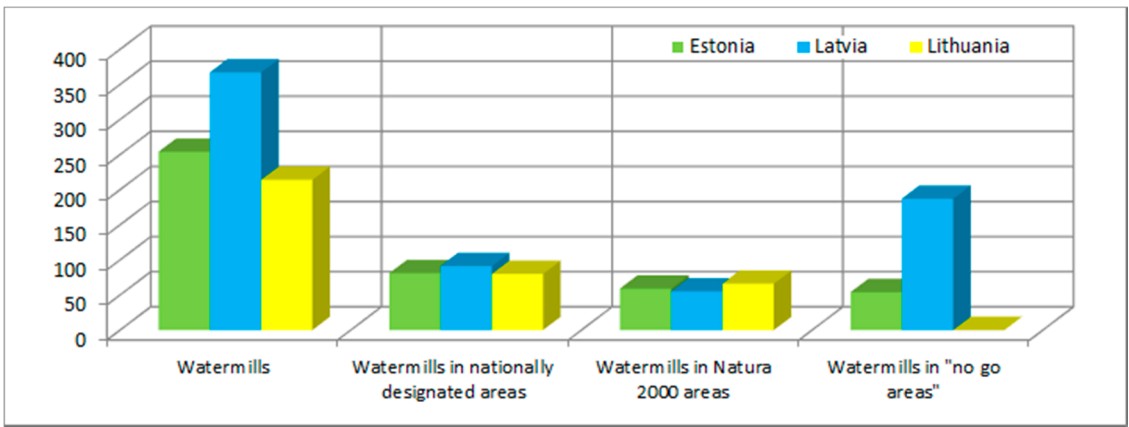

**Figure 10.** The number of watermill sites located in environmentally sensitive areas in the Baltic States.

Almost 22% (180 sites) of watermills in the Baltic States are in Natura 2000 areas. As far as individual countries are concerned, the greatest number of watermills in Natura 2000 areas was identified in Lithuania, followed by Estonia, and then Latvia.

Approximately 30% (252 sites) of all watermills in the Baltic States are located in nationally designated areas. The greatest number of such mills was identified in Latvia, followed by Estonia, and then Latvia.

Finally, almost 29% (241 sites) of watermill sites are located in rivers under restriction ("no go areas"). Overwhelmingly in this case, the leader is Latvia. In Lithuania, after the amendment of the Water Law in 2019, currently, there are no such "no go areas".

In total, the greatest number of watermills located in environmentally sensitive areas is in Latvia (230 sites). This is followed by Estonia (110 sites) and then Lithuania (88 sites). There are multiple watermills identified in overlapping areas, meaning that, in Latvia and Estonia, there are 31 and 21 watermill sites, accordingly, that are located in the area that is a jointly nationally protected area, Natura 2000 area, and "no go area". Similarly, in Lithuania, there are 58 watermill sites that are located in the area that is designated as nationally protected and Natura 2000.

Considerably less—almost 15% (103 sites)—of all existing nonpowered dams in the Baltic States are in environmentally sensitive areas (Figure 11).

The largest number of existing nonpowered dams, almost 11% (75 sites) are in nationally designated areas. Unsurprisingly, as the largest number of such dams were identified in Lithuania, the largest number of them located in nationally designated areas, is also in Lithuania, followed by Estonia, and then Latvia.

Less such dams are in the Natura 2000 areas—7% (50 sites). As far as individual countries are concerned, the leader was again Lithuania, followed by Latvia, and then Estonia.

Only slightly more than 5% of the existing nonpowered dams in the Baltic States are in rivers under restriction ("no go areas"). This number was reduced by the change of laws in Lithuania. Currently, the largest number of dams that are in rivers or river stretches that are in "no go areas" are

in Latvia. Again, the fact that a considerable number of nonpowered dams are in the overlapping environmentally sensitive areas must be taken into account.

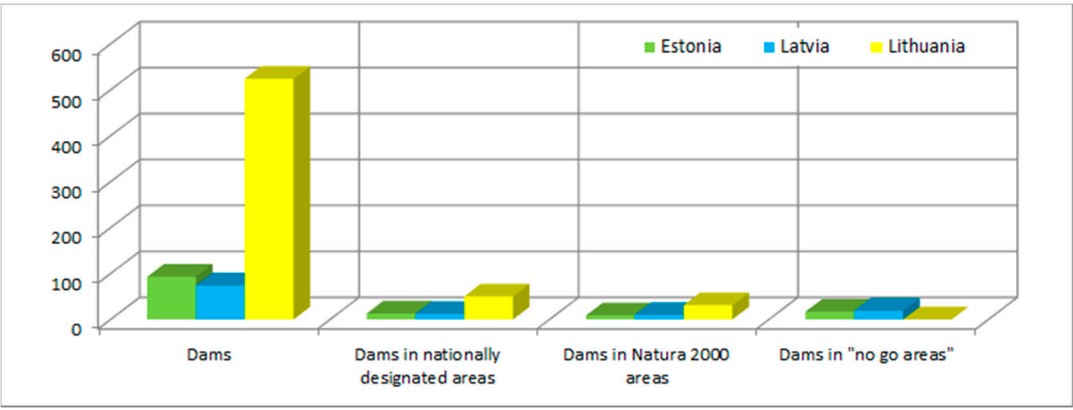

**Figure 11.** The number of existing nonpowered dams located in environmentally sensitive areas in the Baltic States.

A visualisation of the distribution of historic watermills and nonpowered dam sites is presented in Figure 12. This also includes mapping on how many of these sites falls within the boundaries of protected areas in the Baltic States, according to the International Union for Conservation of Nature (Figure 12a) [47], as well as within each category of Natura 2000 sites (Figure 12b) [27] and within rivers or river stretches designated as "no go areas" (Figure 12c). Additionally, historic watermill and nonpowered dam sites that are in overlapping environmentally sensitive areas are categorised as follows: historic sites that fall under one restriction are marked with 1; the sites that fall under two restrictions are marked with 2; and if sites fall under all restrictions considered in this study, they are marked with 3 (Figure 12d). Therefore, in Figure 12d, we present the final view of the distribution of historic watermills and currently nonpowered dam sites that are and are not under restriction for hydropower development in the Baltic States.

3.4.4. Share of the Potential Hydropower Generation of Historic Sites in the Remaining Small Hydropower Potential

This large number of old watermills and nonpowered dam sites in environmentally sensitive areas inevitably decreases the available potential for hydropower development in these historic sites. The share of possible hydropower generation with and without constraints in historic sites in the remaining small hydropower potential is presented in Table 5.

**Table 5.** Share of the historic sites in the remaining small hydropower potential in the Baltic States.

|  | Not Including Constraints | Including Constraints |
|---|---|---|
| **Watermills** | | |
| Estonia | 59.2% | 30.4% |
| Latvia | 46.9% | 6.7% |
| Lithuania | 36.8% | 17.9% |
| **Dams** | | |
| Estonia | 25.4% | 16.7% |
| Latvia | 7.8% | 1.8% |
| Lithuania | 19.9% | 16.5% |

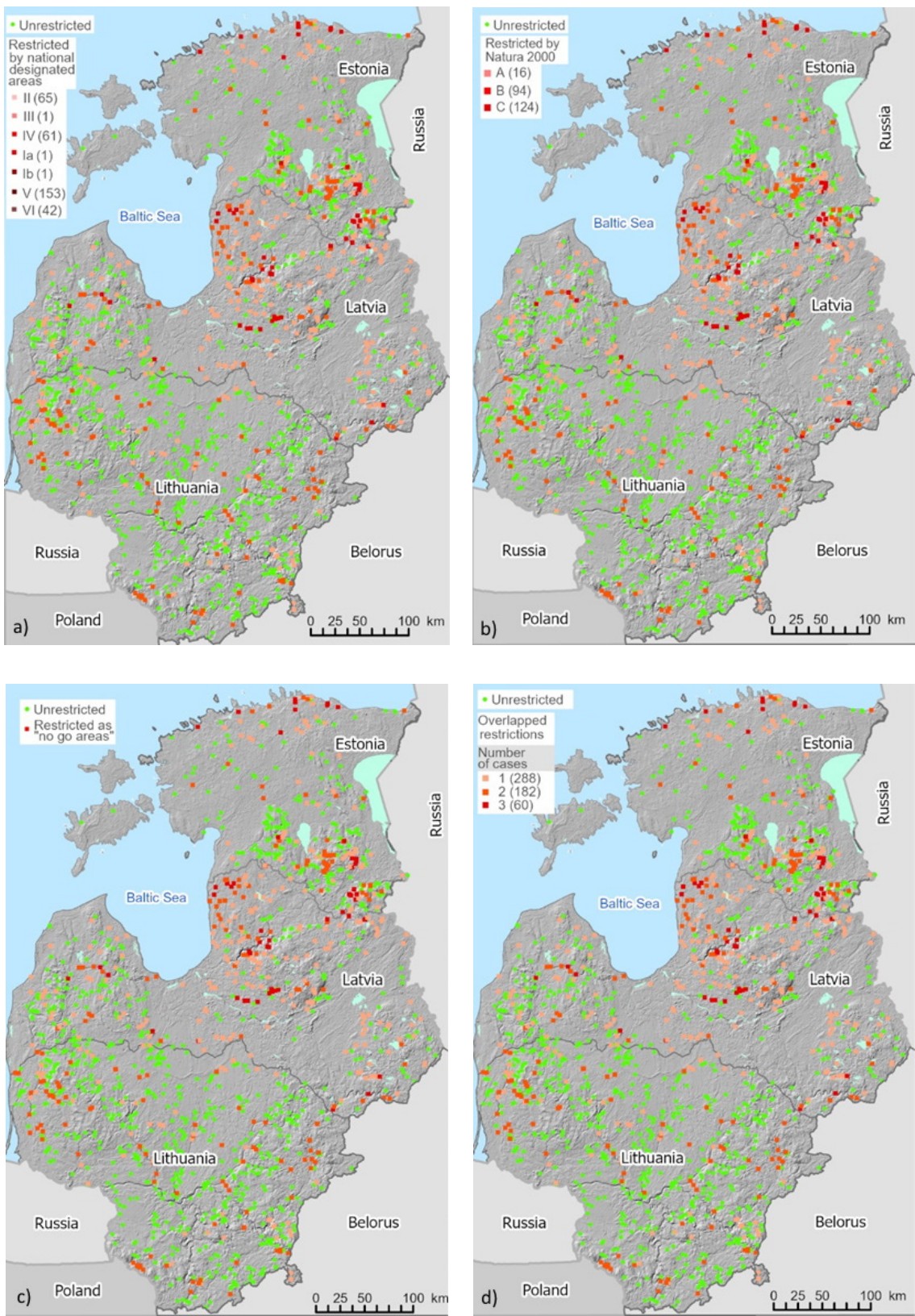

**Figure 12.** Historic and currently nonpowered dam sites in environmentally sensitive areas: (**a**) in nationally designated areas, (**b**) in Natura 2000 areas, (**c**) in "no-go areas" and (**d**) overlapping restrictions (Source: © Esri, © USGS SRTM, open street map contributors). Note: numbers in brackets indicate the number of cases.

Environmental constraints indeed limit the use of historic sites for hydropower development, and the largest limitations were observed in Latvia and Lithuania (Table 5). In total, across all three Baltic States, the possible electricity generation from watermills and nonpowered dams' sites that are located in environmentally sensitive areas was 152.8 GWh/year. This number reduces the share of the historic sites in the remaining small hydropower potential in the Baltic States quite dramatically. The remaining share was 21.6%, which still can be considered a large number. However, for plans to retrofit any of the historic sites for electricity generation, the capacity of the future hydropower plant would be an important factor in the majority of cases. Therefore, the issue would inevitably be that the majority of historic sites with the highest capacity are in environmentally sensitive areas. For example, for the 20 watermill sites with the highest capacity (from 1853 kW to 34 kW) in each country, 18 are in environmentally sensitive areas in Latvia, 17 in Lithuania, and 12 in Estonia. The situation is better with nonpowered dams; however, a considerably larger part of the existing dams is attributed to micro-hydro compared to watermills (Figure 9).

## 4. Discussion

Novelty and originality of this study follow the recommendations of the European Commission first to upgrade existing hydropower plants or use existing in-stream structures for hydropower development. This work has a detailed focus on the most common existing in-stream structures in the Baltic States—historic and currently nonpowered micro-hydro sites. It is building on the state-of-the-art applied science methods, such as a spatial analysis of the potential micro-hydro sites, based on GIS tools.

The history of hydropower in the Baltic States can date back to the 11th century when the first mention of watermills in written sources can be found. Due to various reasons, parts of old watermill sites were abandoned and are not currently used for electricity generation. Similarly, there are a large number of small dams in the Baltic States, built during the Soviet Union era mainly for agricultural purposes, that are also currently nonpowered. Exploiting the current resistance for dam building and impoundment in the European Union, such sites can be proposed for retrofitting and electricity generation. This study demonstrated that there are more than 1500 historic and nonpowered dam sites in the Baltic States.

Overall, there remains a large untapped potential for small hydropower in the Baltic States. We determined that 21.6% (which is approx. 92.3 GWh/year) of this potential can be exploited by retrofitting historic and currently nonpowered dam sites that are not in environmentally sensitive areas.

This study revealed that the overwhelming majority of historic sites in the Baltic States can be attributed to micro-hydro. In scientific studies, small scale hydro is typically considered as "run-of-river" [48] according to the water retention time in upstream storage and its filling period or mode of operation [49,50]. One of the characteristic features of small-scale hydro is low head dams. These dams do not significantly alter the hydrological regime of the river [22,51] and, consequently, have a smaller environmental impact than larger dams, which are usually attributed to large hydropower. Still, there is a high possibility that small dams will also create an unpassable barrier for migrating fish. Retrofitting these dams with hydropower schemes can even improve the environmental conditions as not only fitting these schemes with a fish pass will be mandatory but other environmental measures could also be placed, resulting in a win–win situation [9].

The existing knowledge on retrofitting low head dams also can be of use. Currently, 51% of all small hydropower plants in Lithuania use head that is lower than 5 m. In Latvia and Estonia, this corresponding indicator is 66% and 80%. Only seven SHP plants through all three Baltic States use a head that is higher than 15 m. However, this engineering knowledge is fading. Since 2004, when the law amendments listing protected rivers and their stretches in the Baltic States came into force consequently creating "no go areas", the development of hydropower literally stopped. Moreover, the opposite process is now accelerating. In Estonia, a large project is ongoing to ensure free fish migration in the Pärnu River Basin, where one large dam and six smaller dams will be removed.

The first dam, which was ruins of an old watermill, was removed in Lithuania in the summer of 2020. The second removal in Lithuania is scheduled for 2021. The only country in the Baltic States where the debates on dam removal has not started is Latvia [52].

The historic value of old watermill sites is acknowledged in all three Baltic States, as 36 sites in Lithuania, 32 sites in Estonia, and 122 sites in Latvia (which were considered in this study) are listed as under national heritage protection. The importance of many of these sites is further enhanced by the fact that in these watermills, the original equipment and hydroturbines have often survived to this day. The restoration of these sites can provide a great social boost for local rural communities as watermills can become objects of attraction for tourism and recreational purposes. Already, examples exist in Europe where micro-hydro schemes are owned by local communities [53]. However, even if all possible environmental measures could be enacted at old watermill sites, the retrofitting is prohibited by the environmental regulations in the Baltic States if the watermill site falls into an environmentally sensitive area.

Renewable energy development plans for the Baltic States leave hydropower development behind. This is clearly illustrated by the Lithuanian example. The Lithuanian target for small hydropower development set for 2020 in the national renewable action plan was 40 MW, while same target set for 2030 in the national energy and climate plan is only 27 MW.

Unfortunately, there is little research in the Baltic States to reconcile environmental requirements and small hydro development to clearly support these governmental policies. The already known and emerging state of the art tools (e.g., MesoHABSIM [54], CASiMiR [55], and fish species sensitivity classification [56]) could be adapted and used to assess the environmental impact of low head dams in small and medium lowland rivers in the Baltic States. Even a basin-size assessment for river connectivity purposes can, at the same time, include small hydropower site selection [57], especially while site selection studies are still ongoing even in Estonia, which is the Baltic State with the smallest hydropower potential [58].

The number of hydroturbines, designed for micro-hydro, can be utilised for retrofitting of the historic and nonpowered dam sites [59,60]. It mainly depends on site characteristics, such as flow and head. Proceeding here with caution is mandatory as most historic and nonpowered dam sites in the Baltic States are limited by both. The turbine selection and economic feasibility analysis, using current state supported framework as a basis, are the goals of the follow-up study of this work.

**Author Contributions:** Conceptualisation, E.K. and P.P.; formal analysis, E.K., P.P., A.K. and A.D.; investigation, E.K. and P.P.; methodology, E.K. and P.P.; visualisation, L.J.; writing—original draft, E.K.; writing—review & editing, P.P., A.K. and A.D. All authors have read and agreed to the published version of the manuscript.

**Funding:** This research received no external funding.

**Conflicts of Interest:** The authors declare no conflict of interest.

## Abbreviations

| | |
|---|---|
| CDDA | Common database on designated areas |
| EC | European Commission |
| EU | European Union |
| GIS | Geographic information system |
| NECP | National energy and climate plan |
| NREP | National renewable energy plan |
| PSO | Public service obligation |
| SHP | Small hydropower |

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
