# Peer review of "Small Hydropower in the Baltic States—Current Status and Potential for Future Development"

_energies, doi:10.3390/en13246731_

Round 1

Reviewer 1 Report

06/12/2020

The paper analyses the energy situation in the countries of the Baltic sea, with focus on the small hydropower sector. It gives a wide overview, thus it should be considered from the journal. It is clear and well written.

I did not find errors or substantial lacks.

Anyway, I have some minor comments to be implemented for improving the paper, and two major comments.

Major comments

1) The novelty of the work is not clear. Authors presented the energy situation in three countries, but it is not clear which calculos they did, where is the originality of the work and the scientific value. This article, at the moment, is a very good report, but something of more elaborated should be added to improve the scientific value, and it should be considered for publication in the journal.

For example:

  • how much is the head and flow rate distribution, and their average value? Which kind of turbine would suit in these sites (e.g., X% water wheels, Y% VLH turbines, Z% Archimedes screw). It is not needed to go into too much details, since this would not be the scope of the work being a large scale assessment, but a discussion about this, and few preliminary calculos, should be provided, otherwise this work would not have new scientific insights or novel information. In my opinion, the following classification could be done: for heads below 1.5 m and flow below 3 m3/s per metre width water wheels, for heads below 3 m and flow below 2 m3/s water wheels, for heads below 6 m and flows below 500 l/s  again water wheels, for heads between 2 and 5 m and flow within 0.4- 6 mc/s Archimedes screw, for heads below 5 m and flows above 6 mc/s VLH turbine, or more Archimedes screws. Micro kaplan are excluded due to their high costs. Anyway, for heads above 6 m and flow above 8 mc/s, Kaplan/Francis turbines have to be used, if only a single installation has to be done, otherwise more VLH or screw turbines must be used in parallel, but, at this point, maybe it would be economically better to have a single kaplan than two Archimedes screws. I know that in this large scale assessment it is not possible to enter into these details, but some discussion or general calculos should be done, just to have an order of magnitude, and give suggestions. There may be sites with more possible solutions, and this should be discussed, also in a general way. In this kind of assessment, not too much engineering details are needed, but at least a preliminary and general engineering discussion and data elaboration should be done, considering also the type of Journal. Or a classification based on the available power may be done, e.g. for power below 50 kW water wheels or screws, for power 50-100 kW archimedes screw, for power above 100 kW the VLH turbine. I am always excluding kaplan turbines because of high costs and impacts. I know that in most sites hydro turbines can not be installed due to environmental legislation about which the authors correctly spoke about, but an assessment of "what could be engineering done" may be useful. Some reviews with hydraulic conditions for each turbine are the followings, but authors can implement their own assumptions, mine were just examples.

    - Bozhinova, S., Hecht, V., Kisliakov, D., Müller, G., & Schneider, S. (2013). Hydropower converters with head differences below 2.5 m. Proceedings of the Institution of Civil Engineers-Energy, 166(3), 107-119.

    - Quaranta, E., & Revelli, R. (2018). Gravity water wheels as micro hydropower energy source: a review based on historic data, design methods, efficiencies and modern optimizations, Renewable and Sustainable Energy reviews, (97), 414-427.

  • If the small hydro sites should be retrofitted, how much would be the cost? Speditive equations available in literature can be used, e.g. "Analysis of the cost for the refurbishment of small hydropower plants", with costs as a function of head and power, or simply a cost per installed kW.

2) The article seems to be a scientific review rather than a research article. Please, evaluate the type of article, or better specify.

Minor comments

  • do not use "etc", not scientifically good
  • line 14: which countries?
  • line 20: define "micro"
  • line 44: wind and solar spread would imply an increasing need of energy storage in Europe, i.e. hydropower, see "Assessment of the European potential for pumped hydropower energy storage based on two existing reservoirs"
  • line 45: sentence to be completed I think
  • line 46 and Table 1: which sources? In which sense?
    It would be better to firstly present Table 1, and then comment it with this sentence about the different results from one source to an other
  • line 90: add reference
  • from line 111 to 112: the topic changing from these two paragraphs is to marked: please, discuss why the repowering of old sites may thus be an option in these countries before discussing the scope of the work
  • section 3.1.1 and similar: should this be part of the Introduction? Maybe this work should be considered as a review paper.
  • line 196: how much power?
  • line 240:  water wheels can be used for electricity generation, and in sites within few meters head and low flows, they are the most efficient and cost effective hydropower converter. The reactivation of the water wheel of the mill (after optimization, or replacement of a modern wheel) may also exhibit cultural value.
  • Table 3: better specify each kind of potential and how can you pass from gross to technically feasible, to....and so on
  • Table 4: It would be interesting to show the distribution of heads and flow rates, and their average value
  • Fig.8: this figure is not clear: when water mill sites value is 1, the power is above 100 kW: do you mean that with one water mill you can generate more than 100 kW (considering 2-3 m head?)? This is not possible, maybe it must be better explained. Check also Fig.9
  • lines 446 and 561: but by using existing mill structures, and repowering the water wheel, nothing is substantially changed. Please, better discuss.
  • line 504: better define how it was calculated
  • line 510 :how much power?
  • line 529 :how much GWh/year?

Reviewer 2 Report

The article presents the historical development, the current state and development possibilities of small hydropower plants in Lithuania, Latvia and Estonia. To compare the three countries, the authors used historical data on hydrotechnical structures and data from hydropower associations, national statistics departments and industrial system operators. The structure of the article is preserved, the literature review is selected correctly.

Comments,

The main goals of the work are:
• to present a brief history and overview of the current status of small hydropower in the Baltic States;
• to analyse environmental laws, support schemes, and national renewable energy strategies that  affect the future development of small hydropower in the Baltic States; and
• to evaluate the possible capacity and electricity generation at historic and currently  non-powered dam sites in the Baltic States.

According to the authors, can we treat the history of energy in a given country as the goal of a research task ????

There is not a single formula in a scientific article!!!

Please describe the method used to carry out the historical data analysis. (few sentences),

Please avoid so-called collective quotations, for each quoted article there should be at least two sentences that this article contributed to the guesswork discussed by the authors. For example [18,19]

In my opinion, the article should be bullied as a review, not as a scientific one.

Reviewer 3 Report

The manuscript is interesting. It seems to be more of a review rather than an original article. Nevertheless, it needs some further improvement before being accepted for publication. In general, there are still some occasional grammar errors throughout the manuscript, especially the article ‘’the’’, ‘’a’’ and ‘’an’’ is missing in many places; please make a spellchecking in addition to these minor issues. The reviewer has listed some specific comments that might help the authors further enhance the quality of the manuscript.

  1. Specific Comments

Please include a list of acronyms and abbreviations.

  • Introduction
  • The objectives are not explicitly stated.
  • The authors need to further enrich the background; please highlight the importance of renewable energy sources development in achieving sustainability goals and the importance of the environmental flows in river ecosystem conservation. The following literature might be useful in this regard << Optimisation of cascade reservoir operation considering environmental flows for different environmental management classes>> and <<Water-energy-ecosystem nexus: Balancing competing interests at a run-of-river hydropower plant coupling a hydrologic–ecohydraulic approach>>, << Environmental Flows Assessment in Nepal: The Case of Kaligandaki River>> you may review other additional relevant references as well.
  • What is the novelty of this work?

  • Methods
  • I would suggest showing the methodological approach through a flowchart
  • Methodology limitations should be mentioned.
  • All variables should be explained.
  • Please justify why your model-approach is feasible?
  • What was the data resolution used in the computations?

  • Results
  • This section is well written. Nevertheless, as I mentioned, it looks more like a review rather than an original article.

  • Discussion

The discussion should summarize the main finding(s) of the manuscript in the context of the broader scientific literature and address any limitations of the study or results that conflict with other published work.

The Conclusions section is missing!

Round 2

Reviewer 1 Report

I revised the new manuscript and the Authors well discussed my comments.

I have few minor comments below.

One other general comment is that some of their answers should be included in the manuscript for a better clarity.

Comment 2

Ok, I agree after Author explanation. But maybe they could add such comments in the discussion, in order to stimulate future researches and challenges.

Comment 6

"Not accepted. Small hydropower classification in
terms of capacity will be described in detail in the
Materials and Methods section."

I agree that it will be described later, but, in a paper, every time that a specific term is introduced it must be defined. Therefore, I strongly recommend to define micro hydro the first time that this term is written (I know that micro hydro is < 100 kW, but not all people may know this..)

Line 117: I suggest to add some references with some case studies where historic barriers or non-powered barriers have been retrofitted for hydropower generation, in order to support the Author statements with scientific and engineering evidence. Not necessarily old mills, but any type of unused weir.

Furthermore, I suggest to cite some papers where hydroturbines in this context are used, e.g.

Bozhinova, Snezhana, Veronika Hecht, Dimitar Kisliakov, Gerald Müller, and Silke Schneider. "Hydropower converters with head differences below 2· 5 m." Proceedings of the Institution of Civil Engineers-Energy 166, no. 3 (2013): 107-119.

Vuta, L. I., Dumitran, G. E., Popa, B., Diminescu, M. A., & Tica, E. I. (2019, October). Hidden Hydro Related with Non-Powered Dams in Romania. In 2019 International Conference on ENERGY and ENVIRONMENT (CIEM) (pp. 413-417). IEEE.

Loots, I., Van Dijk, M., Barta, B., Van Vuuren, S. J., & Bhagwan, J. N. (2015). A review of low head hydropower technologies and applications in a South African context. Renewable and Sustainable Energy Reviews50, 1254-1268.

Comment 7

I agree, but I suggest to specify this answer in the manuscript, in order to avoid misleading interpretations.

Lines 150-151: "8 is the coefficient that is the product of the specific weight of water (γ = 9.81 kN/m3) and
the overall plant efficiency (η = 0.6-0.9), Q is the flow (m3/s)"

If the plant efficiency ranges between 0.6 and 0.9, also the coefficient changes. Thus, you selected 8 based on which efficiency? Better specify.

Comment 17.

Please, discuss this in the manuscript, in order to avoid misleading interpretation. Figs.8-9 should thus be better discussed on how to be read.

Comment 18.

Ok, but you should add few lines of descriptions referring to the cited reference.

Table 1

There are 5 columns grouped into 2 categories (2016 and 2020 estimation). It is not clear if the third column refers to 2016 or 2020. Please, better specify how you chose the increase in 1.5 GW.

Line 602

This added reference refers only to water wheels, that are suitable in old mills and in certain hydraulic conditions, where they are the most cost effective  and efficient technology. But, in other conditions (high flows and heads above 2-3 m) water wheels are not the best solution. Thus, I suggest to speak in general about all the possible turbines that can be used (just few lines, not necessary to go into the details, the interested reader can use the references to go into more details), considering the previous suggested references (in particular, I think that Bozhinova et al (2013) is the most appropriate, but the Authors can choose that they prefer and think is the best), and then considering the reference on water wheels specifically for old mills. In my previous manuscript review, I suggested the water wheel reference in addition of Bozhinova et al (2013)  because the Authors spoke also about water mill repowering, and the reference of Quaranta and revelli (2018) on water wheels well suits for water mills. But Bozhinova et al (2013) is more general and should be considered (or any other similar one the Author know....)

Reviewer 2 Report

I accept the authors' answers and explanations.

Author Response

We are very grateful to the anonymous reviewer for investing their own time in reviewing our revised manuscript

Reviewer 3 Report

Acceptable.

Author Response

(The authors gave the same response as above.)
